# pH-Responsive Biomaterials for the Treatment of Dental Caries—A Focussed and Critical Review

**DOI:** 10.3390/pharmaceutics15071837

**Published:** 2023-06-27

**Authors:** Yanping He, Krasimir Vasilev, Peter Zilm

**Affiliations:** 1Adelaide Dental School, University of Adelaide, Adelaide, SA 5000, Australia; yanping.he@adelaide.edu.au; 2College of Medicine and Public Health, Flinders University, Bedford Park, Adelaide, SA 5042, Australia

**Keywords:** pH-responsive delivery systems, pH-responsive release mechanism, tooth-binding, dental caries

## Abstract

Dental caries is a common and costly multifactorial biofilm disease caused by cariogenic bacteria that ferment carbohydrates to lactic acid, demineralizing the inorganic component of teeth. Therefore, low pH (pH 4.5) is a characteristic signal of the localised carious environment, compared to a healthy oral pH range (6.8 to 7.4). The development of pH-responsive delivery systems that release antibacterial agents in response to low pH has gained attention as a targeted therapy for dental caries. Release is triggered by high levels of acidogenic species and their reduction may select for the establishment of health-associated biofilm communities. Moreover, drug efficacy can be amplified by the modification of the delivery system to target adhesion to the plaque biofilm to extend the retention time of antimicrobial agents in the oral cavity. In this review, recent developments of different pH-responsive nanocarriers and their biofilm targeting mechanisms are discussed. This review critically discusses the current state of the art and innovations in the development and use of smart delivery materials for dental caries treatment. The authors’ views for the future of the field are also presented.

## 1. Introduction

Dental caries is a common disease that can occur throughout life and is one of the most prevalent global health problems [1,2]. Although not as intensely studied as other diseases, it affects the vast majority of adults and 60–90% of school-age children in industrialized countries. Thanks to a variety of public health initiatives, including effective fluoride exposure and improved health care practice, dental caries is often well-controlled in developed countries, while an opposite rising trend is observed in low- and middle-income nations. Traditional therapeutic dental care is a significant financial burden in many developed countries, and apart from the economic consideration dental caries can have a negative impact on quality of life, as it leads to pain, the avoidance of food and social embarrassment [2,3,4].

Dental caries is caused by dental plaque, a polymicrobial biofilm that attaches to pellicle, a glycoprotein film that covers all oral surfaces [5]. The key modulators for the development of the cariogenic plaque biofilm are high levels of acidogenic and aciduric bacteria, consisting of members of the mutans Streptococci group which includes *Streptococcus mutans* [6]. *S. mutans* is a highly cariogenic bacterium, capable of fermenting carbohydrates to lactic acid, which initiates the demineralization of the inorganic component of teeth [7,8]. Further, insufficient salivary calcium phosphates and low pH in the oral environment can also contribute to the proliferation of other aciduric and acidogenic bacteria while inhibiting the growth of the beneficial microbiota, ultimately leading to dysbiosis [9,10,11,12]. 

Nonrestorative therapeutic treatments have been widely explored for clinical care, mainly through dental plaque control using antimicrobial agents and remineralisation processes to arrest the loss of dental tissues. These include the use of sodium fluoride (NaF), ammonium fluoride, silver diamine fluoride (SDF), chlorhexidine and silver nitrate, all of which aim to reduce clinical intervention [13,14,15,16,17]. Although reviews of these therapeutic agents indicated that they are effective in arresting caries [18,19,20,21], few innovations have been introduced to improve clinical outcomes and reduce the incidence of caries in the population [22]. 

In recent decades, with developments in nanotechnology, novel pH-responsive controlled drug delivery vehicles have attracted great attention and have been widely developed for biomedicine, especially in treatments for cancer, cardiovascular diseases, hypertension and peptic ulcers [23]. These “smart materials” target detrimental changes in the environment (pH responsive) and produce a controlled release under pathologic conditions, unlike conventional therapy [24]. Similarly, targeted delivery using pH-triggered release technology can be directly applied to the treatment of dental caries. The pH of the localised carious environment can be as low as 4.5, compared to the normal physiological pH range of 6.2 to 7.6 [25]. The main advantage in using these “smart materials” is that they target the dysbiotic biofilm and promote the establishment of the healthy commensal microbiome by selectively reducing the numbers of aciduric bacteria.

This focussed review critically discusses pH-responsive delivery vehicles encapsulated with various therapeutic agents for the treatment of dental caries. It includes preparation and fabrication methods, drug loading processes and innovations in targeting the dysbiotic oral biofilm. We also discuss similar pH-responsive delivery platforms used in other fields, as well as their potential applications in dentistry. 

## 2. Search Strategy

Frequently used databases were searched, including PubMed, Web of Science, Science Direct, and Ovid Medline, for relevant publications with no date restrictions, in accordance with the guidance from the Preferred Reporting Items for Systematic Reviews and Meta-Analyses (PRISMA) statements for methodological quality assessment [26]. Keywords used for the search in all databases were “pH-responsive”, “pH-sensitive” and “dental caries”. pH-responsive/sensitive delivery vehicles for dental caries treatment were the main focus for data and information extraction, and any subjects not relating to this were excluded. Research articles with keywords in the abstracts and content focusing on the topic of this review were included. An additional plain Google search was performed using the above-mentioned selection criteria (see Figure 1). Pertinent references from the bibliographies of the consulted articles were also examined for applicable references that were not shown by the database and also to ensure an effective follow up of the relevant studies for the present review. 

## 3. Nanoparticle Formulation and pH-Responsive Release Mechanism 

pH-responsive nanocarriers are the most widely used drug delivery systems in the oral cavity. To enable pH-responsive release, nanocarriers are generally fabricated with pH responsive groups such as amines or acid-labile bonds. pH changes induce the protonation/deprotonation or lysis of the chemical bonds, resulting in drug release [27]. Various polymers are used for the preparation of nanocarriers, and the following sections categorise nanocarriers by their polymer composition.

### 3.1. DMAEMA

DMAEMA (dimethylaminoethyl methacrylate) is a cationic monomer and its polymer (p(DMAEMA)) has often been used as a drug delivery vehicle due to its pH-responsive properties. The chemical structure (Figure 2) of DMAEMA contains a tertiary amine group which has a pKa of 7.5. It undergoes structural changes (swelling) in acidic conditions due to the protonation of the tertiary amino moity, and becomes hydrophilic. When the environmental pH is above its pKa, the amine group will be deprotonated and becomes hydrophobic [28,29], which is an important property in loading hydrophobic drugs.

Horev et al. (2015) and Zhou et al. (2016) used a two- step reversible addition–fragmentation chain transfer (RAFT) polymerization method to synthesise a delivery vehicle which allowed the modulation of the polymer molecule weights and polydispersity indices (PDI) [30,31]. In the first step, poly (dimethylaminoethyl methacrylate) p(DMAEMA) was synthesised by introducing DMAEMA and dimethylformamide (DMF) into a reaction vessel with a chain transfer agent (CTA). The initiator (2,2-azobisisobutyronitrile) was then added, and the polymerization reaction occurred at 60 °C for 6 h. In the next step, p(DMAEMA) was used as a macro CTA and crosslinked with DMAEMA, 2-propylacrylic acid (PAA), and butyl methacrylate (BMA) monomers (25:25:50%, respectively) to form p(DMAEMA)-b-p(DMAEMA-co-BMA-co-PAA) diblock copolymers with 2,2-Azobisisobutyronitrile (AIBN) as the initiator. The copolymers were then self-assembled into micelles due to the hydrophobic interactions among BMA residues in the p(DMAEMA-co-BMAco-PAA) core blocks.

The loading of farnesol inside the micelles was achieved via sonication, with a loading ratio up to 22 wt% and a loading efficiency of 100%. Farnesol is a hydrophobic antibacterial agent, but has limited antibiofilm effects after topical applications. The pH-responsive delivery vehicle was therefore used to enhance drug efficacy through high affinity binding and pH-responsive drug release. The farnesol release half-life was 7 h and 15 h at pH 4.5 and pH 7.2, respectively, indicating pH-responsive release behaviour. Moreover, nearly all farnesol (75%) was released within 12 h at pH 4.5, in contrast to the 30 h required for complete release at pH 7.2. At physiological pH (pH 7.2), DMAEMA was protonated and PAA was deprotonated, creating an amphiphilic core for farnesol, while at an acidic pH (pH 4.5) DMAEMA was completely protonated and PAA was neutralized. Electrostatic repulsion occurs due to protonation between the micelle corona and the core, leading to the disruption of micelle structures and drug release (see Figure 3). Significant antibiofilm activity of farnesol-loaded nanoparticles was observed, with an 80% decrease in the *S. mutans* biofilms’ viability compared to a modest 20% reduction in the free farnesol treated group. In addition, nanocarriers by themselves did not show any antibacterial/antibiofilm activity. The enhanced antibiofilm effect was likely due to the greater binding between the positively charged nanoparticles at acidic conditions and the negatively charged microbial surfaces, as well as the pH-triggered drug release, which prolonged the farnesol retention and increased drug bioavailability at the pellicle–biofilm interface as well as inside the biofilm [30,31].

Peng et al. (2022) also used DMAEMA to synthesise poly(DMAEMA-co-HEMA) to encapsulate chlorhexidine (CHX); the overall structure was known as p(DH)@CHX, and the loading capacity and encapsulation efficiencies of CHX in poly(DMAEMA-co-HEMA) were 16.03% and 80.15%, respectively. When placed in an acidic environment, the amine groups in the DMAEMA polymer become protonated, resulting in structural changes (swelling) and causing the subsequent release of CHX. Results from the release study confirmed that p(DH)@CHX released CHX significantly faster in the acidic environment than in the neutral environment, with 60% of CHX released in the acidic condition compared to 30% in pH 7.4 after one hour [29]. CHX is considered a gold standard antiplaque agent (mouth rinse) in dental clinical practice and has been widely used for its broad-spectrum antibacterial effects [32]. Nevertheless, the long term application of CHX is not recommended due to several disadvantages, including cytotoxicity, and few extenuating effects on oral biofilm [33,34,35]. In the study, the p(DH)@CHX retained the same antibacterial/antibiofilm effect as free CHX, and both reduced lactic acid production and biofilm viability to 80%, respectively, compared to the control, yet no significant improvement was credited for using the delivery vehicle, except for a lower cytotoxicity. Therefore, more efforts are required to optimize the current delivery system and to enhance adhesion to the biofilm [36]. 

### 3.2. Polyethylene Glycol (PEG)

Poly (ethylene glycol) (PEG) is the one of the most frequently used hydrophilic polymers used in drug delivery and nanotechnology. Considered a gold standard with “stealth” properties, it has fewer interactions with blood components, thus avoiding the rapid blood clearance seen with other drug carriers, and has excellent biocompatibility [37,38]. Furthermore, PEG can be used to stabilize polymeric nanocarriers as it decreases the chance of agglomeration of the particles via steric stabilization, increasing the stability of the produced formulations during storage and application [39,40]. In addition, PEG is highly soluble in organic solvents, making modification in the end-group comparatively simple, and it is also suitable for biological applications because of its water solubility and low intrinsic toxicity [37]. Due to these superior properties, self-assembled micelles using poly (ethylene glycol) (PEG) as the corona-forming block have been investigated further [41], as they can be modified with pH-responsive groups or acid-labile bonds which respond to low pH, causing structural changes that release the drug. Relevant research regarding self-assembled micelles using PEG as the corona-forming block for pH responsive drug delivery systems is listed in Table 1.

Zhao et al. [42] synthesised a cationic poly (ethylene glycol)-block-poly(2-(((2-aminoethyl) carbamoyl) oxy) ethyl methacrylate) (PEG-b-PAECOEMA), then modified with citraconic anhydride (CA) to form negatively charged PEG-b-PAECOEMA/CA as the pH-responsive delivery vehicle for CHX. PEG-b-PAECOEMA/CA could encapsulate cationic CHX via electrostatic interactions and self-assemble into core–shell polyionic complex micelles (PICMs). In this structure, the neutral PEG block was used for the stabilization of the polymer and the PAECOEMA/CA block as the pH sensitive group. In neutral conditions, the PAECOEMA/CA block was negatively charged because of the carboxylate groups at the end, facilitating the encapsulation of the cationic CHX inside the core via electrostatic interactions, and exposing PEG as the shell. The loading efficiency and encapsulation efficiency of CHX in CA-PICMs were detected at 16.48% and 75.02%, respectively. In a healthy oral microenvironment, the synthesised PICMs were relatively stable, but in acidic environments, citraconic amide was degraded and transformed to a positively charged primary amine (Figure 4), facilitating the rapid release of CHX from the micelles via electrostatic repulsion. Nearly 69% of CHX was released from CA-PICMs in the first 3 h, compared to 39% at pH 7.4 during the same interval, confirming the pH responsive release profile of CHX from PICMs. Furthermore, they also demonstrated the pH-responsive bacterial killing of CA-PICMs on *S. mutans* biofilms grown on hydroxyapatite (HA) discs, which was represented by an abundance of dead bacteria (dyed red) after the treatment with CA-PICMs via live/dead bacterial staining, with a similar dead/live bacteria ratio to the CHX group. Although no antibacterial effect was found in the delivery vehicle (PEG-b-PAECOEMA/CA), CA-PICMs were less cytotoxic to human oral keratinocyte (HOK) cells compared to the corresponding free CHX, which greatly reduced the negative impact of CHX on oral tissues.

Zhang et al. (2021) used mPEG-b-PDPA to produce a pH-responsive core–shell nano micelle [44]. 2-(Diisopropylamino) ethyl methacrylate (DPA) copolymer was used to form the hydrophobic core for loading bedaquiline (a hydrophobic antibacterial agent). The loading ratio and encapsulation efficiency of bedaquiline in mPEG-b-PDPA were 37 and 92.5%, respectively. The micelles acted as a pH-responsive agent that shifted the core from hydrophobic to hydrophilic via protonation as the pH dropped below 6 (see Figure 5). This caused the swelling and disassembly of the micelles and subsequent drug release. At pH 5, the cumulative release of bedaquiline from micelles was 92.2% in 3 h, but no more than 35% was detected at pH 7 in the first 12 h. This nanocarrier not only enabled the targeted release of the therapeutic agent under acidic environments, it also improved the working concentration in the local acidic area to exert a greater antibacterial/antibiofilm effect. In the antibacterial model at pH 5, free micelles did not show an inhibitory effect against planktonic *S. mutans*, while the 1% bedaquiline-loaded micelles group (equivalent to ~25 μg mL^–1^ bedaquiline) significantly reduced the growth of *S. mutans* to less than 50% of the control. In addition, the live/dead staining images of the *S. mutans* biofilm revealed that most bacteria in the biofilm were dead (dyed red) in the 1% bedaquiline-loaded micelles group, as opposed to the massive green cells observed in the control and free micelle groups. Cytotoxicity studies using periodontal ligament stem cells showed that the micelles were not toxic, suggesting a good potential for clinical application for the treatment of dental caries. 

PEG-based polymers can also be modified with acid liable bonds, which are cleaved upon changes in pH and release therapeutic agents. For example, Xu et al. (2023) [45] developed MAL-PEG-b-PLL/PBA-based polymeric micelles (PM) and loaded them with sodium fluoride (NaF) and tannic acid (TA) for the treatment of dental caries. In this nano-construct, the phenylboronic acid group in PBA forms a pH-sensitive boric acid ester bond with tannic acid via phenylboronic acid-catechol interactions, which can be cleaved under acidic (cariogenic) conditions. Poly(l-lysine) (PLL) is positively charged at physiological pH and interacts with TA (negatively charged) to form core–shell structure micelles. NaF was co-loaded during the micelles’ assembly. The loading content (LC) and loading efficiency (LE) of NaF in the nanocarrier were 5.5% and 20.9%, respectively. For TA, the LC and LE were 8.3% and 31.7%, respectively. The pH-responsive cleavage of the boronate ester enabled a controlled release of TA and NaF, the release rate for which were 70% and 80%, respectively, within 24 h at pH 5.0, much faster than those at pH 7.4 (45% and 50%, respectively). The constructed nanoparticles (PMs@NaF-SAP) were also coated with the salivary-acquired peptide DpSpSEEK (SAP) to enable selective binding to the tooth enamel surface. The pH-responsive antibacterial/antibiofilm effect of PMs@NaF-SAP was comparable to CHX (ca. 25% of reduction in biofilm formation compared to control), but with less cytotoxicity, and produced minimal changes in the oral microbiota diversity. 

Yi et al. (2020) exploited acid-labile bonds as a pH-responsive release mechanism for PPi-PEGhyd-Far polymeric micelles [47]. Farnesal (Far) is a hydrophobic derivative of farnesol and has proven anti-caries efficacy [48,49]. To increase the therapeutic bioavailability, Farnesal was linked to PEG via an acid-labile hydrazone bond to form PEG-hyd-Far for enhanced solubility, which was then further conjugated to pyrophosphate (PPi) and self-assembly to form PPi-Far-PMs for enamel-targeting delivery. The HPLC method was utilised for determining the Far loading and encapsulation efficiencies in PPi-Far-PMs, which were 9.51 ± 0.40% and 78.30 ± 1.40%, respectively. The pH responsive release profile of PPi-Far-PMs was confirmed, with 90% of Far being released within 24 h at pH 4.5 in contrast to 40.6% at pH 7.4. Additionally, the in vivo anti-caries study showed that blank PMs had no antibacterial effect, while PPi-Far-PMs significantly reduced the amount of *S. mutans* compared to CHX (less than 30%). The collective evidence suggested that this delivery system could be used for the targeting delivery of antibacterial agents in the oral environment, and also as a potential treatment or prevention tool for dental caries.

### 3.3. Chitosan

Chitosan (CS) is a copolymer that is produced by deacetylating chitin in the presence of alkaline chemicals [50]. It is the only known naturally occurring polycationic polysaccharide that can form complexes with anionic molecules [51,52,53] due to its biodegradability and biocompatibility. These features have inspired innovative nanotechnology and glycol-chemistry to produce CS-based nanoparticles as a promising drug delivery vehicle in biomedical and pharmaceutical industries [54,55,56].

Nguyen et al. (2017) reported the preparation of stable, spherical, and monodisperse CS nanoparticles for loading with NaF using sodium tripolyphosphate (TPP) as a crosslinker. The loading capacities of fluoride for chitosan nanoparticles prepared in 0.2% NaF and 0.4% NaF were determined at 33 and 113 ppm (μg/g), respectively, and the corresponding entrapment efficiencies (%EE) were 3.6 and 6.2%, respectively. A high level (55% at pH 5 and 43% at pH 7 at 24 h) of fluoride release was observed at pH 5 compared to pH 7 [57]. The same method was applied by Zhu’s group for the preparation of histatin (HTNs)-loaded CS nanoparticles, which was based on their transition from liquid to gel via ionic interactions with a polyanion [58]. Briefly, histatin (HTN3), which has demonstrated properties in tooth homeostasis and dental caries prevention, was first mixed with CS solution. The mixture then went through ionic gelation with TPP to form HTN3-loaded CS nanoparticles. Results showed that the loading ratio of HTN3 in the CS nanoparticles was tunable, without altering the particle size or dispersity. And the pH responsive release profile of CS nanoparticles was confirmed, as they selectively swelled under acidic conditions, which accelerated the release of HTN3 (58% ± 9% at pH 4 and 2% ± 2% at pH 6.8). However, CS nanoparticles loaded with or without HTN3 showed a significant reduction in bacterial viability (by half) compared to the control and almost the same antibiofilm effects against *S. mutans* with a lower biofilm mass. Specifically, the average wet biofilm mass was detected at a descending trend, i.e., 15 ± 2 mg for the control group, 12 ± 1 mg for HTN3, 8 ± 2 mg for fluoride, 7 ± 1 mg for unloaded CNs, and 6 ± 1 mg for HTN3-loaded CNs. This result indicated the bacterial inhibitory effects of CNs by themselves, and no significant contribution of HTN3 was observed for the antibiofilm/antibacterial activity. Even though CNs can be considered a potential nanocarrier, future studies could consider optimizing the mass ratio between CNs and HTN3 to increase the loading amount of HTN3 in CNs for a better synergistic antibacterial/antibiofilm effect.

### 3.4. Mesoporous silica Nanoparticles

Mesoporous silica nanoparticles (MSNs) are a group of inorganic porous materials that have been extensively studied as a drug delivery platform since 2001 [59,60,61,62]. This is due to their favourable properties, including tuneable pore size, large pore volumes that allow for high cargo loading, and high specific surface area, which facilitates surface functionalization [63,64,65,66,67]. Through surface modification, MSNs can be engineered to be stimuli-responsive controlled release systems, with the release of medications triggered by intracellular stimuli or the alterations in the microenvironment of the diseased sites [68,69,70,71]. For the treatment of dental caries, MSN-based pH-responsive delivery is of special interest as a stimuli-responsive system.

Akram et al. (2021) synthesized poly-L-glycolic acid (PLGA)-grafted MSNs and incorporated chlorhexidine (CHX) [72]. A typical method using cetyltrimethylammonium bromide (CTAB) as a structure-directing agent/template and tetraethylorthosilicate (TEOS) as the silica source was used for synthesis. This was followed by functionalization with L-glutamic acid γ-benzyl ester (BLG) on the surface to form PLGA-grafted MSNs (MSN-PLGA). CHX was then incorporated into MSN-PLGA via simple immersion, sonication, and centrifugation. The loading and encapsulation efficiency were 24% and 96%, respectively. A pH-dependent release of CHX from the CHX-loaded/MSN-PLGA was observed, and the cumulative release of CHX reached up to 70% at pH 5.0 after 24 h, substantially higher than that at pH 7.4 (49%). The difference in release profiles was attributed to the tendency of both MSN and PLGA to fracture at low pH [73]. The antibiofilm effect was evaluated using an MTT assay, and the best performance was demonstrated by the 50:50:50 CHX-loaded/MSN-PGA-treated group after 24 h, for which the *S. mutans* biofilm viability was decreased to less than 20% compared to 90% in the MSN-PLGA treated group. In addition, the biofilm viability remained lower than 30% even after 30 days, indicating a continuous release of CHX at a low pH. All experimental nanoparticles demonstrated low cytotoxicity profiles, with the viability levels of the treated dental pulp stem cells (DPSCs) all above 80%. Although the current results suggested that the MSN-PLGA was promising as a pH-responsive nanocarrier delivery system for CHX, further research is recommended to include multi-species bacterial biofilms to confirm antimicrobial activity. In addition, future research will also investigate whether resin infiltration has an impact on the release profile of the nanoparticles after mixing with commercial dentin adhesive systems. 

Fullriede et al. (2016) developed poly(4-vinylpyridine) (PVP)-modified nanoporous silica nanoparticles (NPSNPs) for the delivery of CHX [74] and tested their antibacterial activity against *S. mutans* and Staphylococcus *aureus* (*S. aureus*). The methodology for loading CHX into a nanocarrier was via the simple incubation of the CHX solution with nanoparticles for 3 days, and D-gluconic acid was used to adjust the pH to 3 to open the pores. The PVP-modified NPSNPs were able to incorporate 24 wt% CHX. PVP also served as a pH-responsive gatekeeper, as it blocked the pore openings under physiological pH and prevented the release of CHX. However, in acidic conditions (e.g., bacterial infection), the PVP polymer chains became protonated and repelled each other due to electrostatic repulsion, leading to opened pores and CHX being released (Figure 6). This assumption was later confirmed by the cumulative release of CHX at pH 4 compared to pH 7.4. A pH-responsive burst release of CHX occurred in the first 12 h at pH 4 and reached 260 μg mg^−1^; this was higher than the 220 μg mg^−1^ released at pH 7.4. Following the burst release, a constant release of small amounts of CHX was observed at both pH 4 and 7. Antibacterial efficacy was performed on *S. mutans* and *S. aureus* planktonic suspensions as well as biofilms, and the results showed a strong antibacterial effect against both planktonic suspensions at higher concentrations than 5 μg mL^−1^, reducing the bacterial viability to less than 20%, while no obvious antibacterial effect was observed for the delivery vehicle. However, no antibiofilm effect was found against the corresponding mature biofilms across all experimental concentrations of CHX-loaded NPSNPs. The cytotoxicity study of gingival fibroblasts revealed that the cytocompatibility of NPSNPs loaded with CHX was up to 25 μg mL^−1^. Although the biological evaluation of CHX-loaded NPSNPs provided a therapeutic window where fibroblasts were still viable at a bacterial inhibition concentration, more optimization work is still required. For example, an increased loaded amount of CHX in combination with a better chemical crosslinker to compact the polymer corona may explain the poor antibacterial effect against biofilms and the unwanted release of CHX at physiological pH. 

To achieve maximum antibacterial efficacy and given that the oral biofilm was characterized for locally acidic and reducing microenvironments, Lu’s group (2018) incorporated redox-active disulfide bonds into the mesoporous silica nanoparticle (MSNs) framework for the dual delivery of both silver nanoparticles and CHX [75]. The developed Ag-MSNs had a large pore size and therefore were able to load higher levels of CHX (21.5% ± 2.2%). The release of CHX and Ag ions happened in a GSH- and pH-responsive manner, where 55% of CHX and 0.5 ppm of Ag ion were released at pH 5.5 with 5 mM of GSH after 24 h compared to 15%, and 0.15 ppm at pH 7.4 without GSH. This supported the assumption that the disintegration of the Ag-MSN matrix consisting of disulfide bridges was accelerated under acidic and reducing conditions. *S. mutans* biofilm was used as a model for the investigation of the antibacterial/antibiofilm activity using Ag-MSNs@CHX, the minimum inhibitory concentration (MIC) was 12.5 µg/mL, and the minimum bactericidal concentration (MBC) was 25 µg/mL. For the antibiofilm test, the Ag-MSNs@CHX group demonstrated the highest biofilm inhibition, with biofilm viability reduced to around 20% at the minimal biofilm inhibitory concentration (MBIC) (50 µg/mL) compared to the control (100% biofilm viability) and Ag-MSNs groups (50%), and the inhibition occurred over 72 h. This indicated a strong synergistic antibiofilm effect due to the continuous release of CHX and silver ion over the long term. Cytotoxicity studies with human immortalized oral epithelial cells (HIOECs) showed that the encapsulation of CHX in the Ag-MSNs nanocarrier could significantly reduce the toxicity of CHX, although the Ag-MSNs@CHX-treated group only retained 60% cell viability after 6 h incubation. 

### 3.5. Tertiary Amine Modified Restorative Resin

Secondary caries has frequently been observed in resin composite restorations [76,77], characterized by resin surface degradation which increased surface roughness and decreased hardness; at the same time, the unpolymerized monomer and dentin binding agent were eluted from composites, promoting the growth of cariogenic microorganisms [78,79,80,81].

To address this problem, Liang et al. (2020) synthesised two monomers based on tertiary amine (TA), DMAEM (dodecylmethylaminoethyl methacrylate) and HMAEM (hexadecylmethylaminoethyl methacrylate), and incorporated them into an adhesive resin to make TA-modified resins (TA@RAs). TA@RAs showed antibacterial effects only in an acidic environment, as the nitrogen atoms of TA were protonated in low pH and formed quaternary ammonium monomers (QAMs) for bacterial killing [82,83]. The pH-responsive antibiofilm effect of TA@RAs was observed against *S. mutans* biofilms, with bacterial viability significantly decreased to less than 20% at a low pH compared to that at a pH above 5.5 (more than 70%). 16S rRNA gene sequencing results showed significantly higher microbial diversity in the DMAEM/HMAEM group than in the control group, which further confirmed the pH-responsive antibacterial effect of TA@RAs, indicating the potential of TA@RAs in preventing secondary caries and shifting the oral microbial community towards a healthy and balanced condition [84].

## 4. Innovation in Binding to Tooth Surfaces and Extracellular Matrix

Dental caries is caused by cariogenic bacteria, which exist and benefit from the development of biofilms (dental plaque) that cover the pellicle-coated tooth surfaces and produce acid through carbohydrate fermentation, resulting in the demineralization of tooth enamel [85,86]. The biofilm extracellular polymeric matrix provides protection to bacteria from environmental changes such as pH, osmolarity, mechanical and shear forces [87,88,89]. It also restricts the penetration of antibiotics, making sessile bacteria more resistant to antibacterial compounds compared to planktonic bacteria. This opens the possibility for bacteria to develop antimicrobial resistance [90,91]. In addition, the ineffectiveness of many common antimicrobial agents against cariogenic biofilms is attributed to the insufficient retention time of the therapeutic agent on the topically applied dental surfaces, and poor drug bioavailability within the biofilm matrix [92,93,94].

Therefore, the key to improving the antibiofilm efficacy is to increase the retention time and maintain a high working concentration of the therapeutic at the “at-risk” area. This can be achieved by the effective binding of the nanocarrier to the biofilm matrix. Furthermore, the acidic pH of the cariogenic biofilm can also be utilised for better binding, micelle disintegration and pH responsive drug release [95,96,97]. Other factors, such as polymer size, ionic strength, charge density, etc., may also be explored for surface binding [98,99,100]. In addition, biomolecules that bind to specific sites, such as the proteins in pellicle or carbohydrates/glycosyl links in the biofilm matrix, can be considered for improving binding [101,102,103,104]. 

Cationic diblock co-polymer nanoparticles such as p(DMAEMA) were reported to have a high binding affinity to tooth surface and glucose-coated biofilm surfaces due to surface tertiary amine residues [30,31,105]. Horev et al. (2015) produced micelle nanoparticles which consisted of cationic poly (dimethylaminoethyl methacrylate) (p(DMAEMA)) coronas and pH-responsive p(DMAEMA-co-BMA-co-PAA) cores. The demonstrated a high binding affinity to the negatively charged tooth surface (hydroxyapatite and pellicle), and exopolysaccharides (EPS) via electrostatic interactions were attributed to the multivalent binding of tertiary amines of p(DMAEMA) [30]. Basically, the absorption of the p(DMAEMA) nanoparticles by hydroxyapatite (the mineral component of tooth enamel) was dependent on the protonation degree of the tertiary amine residues of the p(DMAEMA) corona. The binding was strong at the acidic condition (pH 4.5), as the DMAEMA residues (pKa~7.2) were fully protonated and bound effectively to the negatively charged pellicle, the bacterial membrane and the biofilm surface [93,106,107,108,109]. At the physiological condition (pH 7.2), half of the DMAEMA residues were protonated and attached the nanoparticles to the negatively charged sites of the dental surface via electrostatic attraction [107,108,109,110]. In contrast, DMAEMA residues were deprotonated at a high pH (pH 10.5), and thus no binding to HA (0.5%) was found and only a small amount were detected on the saliva-coated HA (25.9%) and Gtf-derived EPS (glucans)-coated HA (36.2%). Overall, the protonation dependent binding capability of p(DMAEMA)) coronas makes it a desirable targeting moity for the low pH cariogenic sites, and a suitable delivery construct when used in combination with the pH-responsive p(DMAEMA-co-BMA-co-PAA) core for existing or novel antimicrobial agents’ delivery. And the application can be extended beyond mouth to other biofilm-related infections.

Targeted binding to hydroxyapatite, which is the major component of tooth enamel, could be a highly desirable technique to prevent rapid clearance and extend the residency duration of the drug in the oral cavity. Xu et al. (2023) [45] conjugated the salivary-acquired peptide sequence-DpSpSEEK (SAP) [111,112] to nanoparticles (PMs@NaF-SAP) for selective binding to tooth enamel and to tolerate the buffering effect of saliva in the oral environment. Generally, the orally exposed tooth surface is covered by the salivary pellicle, which consists of a variety of salivary proteins. Among these proteins, statherin in particular is in high abundance with calcium-binding domains that bind selectively to HA within seconds or minutes, leading to the rapid formation of pellicles on the tooth surface [111,112,113,114]. And the predominant domain of statherin immobilised on the HA surface is the N-terminal hexapeptide sequence DpSpSEEK (SAP) [115]. This peptide sequence, after conjugating to PMs@NaF, provided the nanoconstruct PMs@NaF-SAP with strong tooth enamel adhesion. In combination with the pH responsiveness of PMs, PMs@NaF-SAP significantly enhanced the efficacy and retention time of NaF and TA in the oral cavity and exerted effective antibiofilm effects without changing the oral microbiota diversity or cytotoxicity on cells and tissues. All in all, this developed smart delivery system can be used for other antimicrobial agents (or co-loaded in PMs@NaF-SAP) and translated to clinical products such as mouthwash, spray or paint for caries prevention, treatment and restoration therapy.

Other moieties such as pyrophosphate (PPi) [116,117] were conjugated to PEG-hyd-Far [47] polymeric micelles (PMs) to mediate the rapid and effective adherence of the PM to tooth enamel. PPi is a biodegradable tooth-binding moiety. Owing to its high affinity for enamel, dentin and tartar [118], it has been widely used in dental care products for abrasion, whitening and anti-tartar activity [119]. Yi’s group (2020) linked PEG with the hydrophobic antimicrobial agent (Far) via an acid-labile hydrazone bond to make PEG-hyd-Far, for enhancing the solubility and pH-responsive release of Far in low pH environments [47]. And they further modified it with PPi to endow the PPi-PEGhyd-Far polymeric micelles with enamel-targeting capability. The results showed that PPi-Far-PMs bound more efficiently to hydroxyapatite than Far-PMs, and persisted for up to 12 h. The anti-caries efficacy of Far was significantly enhanced, suggesting that this drug delivery platform was suitable for routine use in dental caries management.

## 5. Lessons from Other Fields That Could Be Adopted in Dentistry

pH-responsive drug delivery systems are considered “smart” as they are designed to deliver therapeutic agents in a controlled manner and release in response to pH changes, which overcomes the drawbacks of conventional drug formulations and increases therapeutic efficacy [120]. They have been extensively used in a wide variety of non-dental applications [121,122], especially in the field of wound healing [123,124,125] and cancer therapy [126,127,128]. The results are due to the pH levels of some malignancies, the inflammatory tissue and the wounded area being different from healthy tissue. Infected wounds have been found to have a pH above 7, and the value may reach up to 9.6 depending on the wound severity, while healthy skin is slightly acidic, with a pH between 4 and 6.3 [124,129,130]; thus delivery systems have been engineered to release antimicrobial agents at a high pH to maximize the therapeutic efficacy. On the other hand, cancer tissue is reported to be slightly acidic (pH < 6.5) compared to healthy tissues [131,132], and a low pH is also found in intracellular compartments [133,134], such as the early endosome (pH ≈ 5.5–6.0) and lysosomes (pH~4.5–5.0), which can be exploited for the intracellular release of anticancer drugs [135,136]. Since this review had a central focus on dental caries treatment which targets a responsive release in low pH conditions, the delivery systems used in cancer therapy may be adaptable. Recent examples of commonly used pH-responsive delivery vehicles used in cancer therapy are summarized in Table 2; only a few examples have been selected from of a large number of publications and discussed as a potential design reference in dentistry.

Biodegradable poly (lactic-co-glycolic acid) nanoparticles (PLGA NPs) are widely used as drug delivery vehicles. To avoid the premature release of anticancer drugs in healthy tissue, Hu et al. (2020) developed the tannic acid−Fe (III) complex-modified PLGA nanoparticle platform (TPLGA NPs) for the delivery and pH-responsive release of doxorubicin (DOX) [162]. DOX was loaded inside the PEGylated-PLGA inner core with an encapsulation efficiency of 62.84 ± 7.76%, and then coated with an Fe (III)−TA complex outer shell. At low pH conditions, the hydroxyl groups in the Fe (III)−TA complex became protonated, which resulted in the rapid disassembly of the coated shell, which further sped up the hydrolysis of ester bonds in the PLGA polymer, leading to the accelerated release of DOX. In high pH conditions, Fe (III)−TA maintained its tricomplex structure and served as the coated shell, slowing down the hydrolysis of the polymer and the subsequent DOX release [163,164]. Moreover, using an MTT assay, blank TPLGA NPs were nearly non-cytotoxic against breast cancer cells, indicating that the pH responsive delivery platform could be a promising method to enhance the safety and efficacy of therapeutic agents. It is noteworthy that tannin acid (TA) in the nano-platform had a high binding affinity to both the pellicle layer and free salivary enzymes [165,166], which could be used for extending the residential time of the delivered nanoparticles in the oral environment and blocking specific receptors for bacterial attachment [167,168].

Hsu’s team synthesised hydrophilic PEGylated chitosan segments which were conjugated to hydrophobic 4-(dodecyloxy)benzaldehyde (DBA) molecules via the formation of acid-sensitive benzoic-imine bonds [145], which self-assemble in an aqueous solution to form core-shell polymeric micelles (PMs). Indocyanine green (ICG) was used as a model drug, and was encapsulated into the hydrophobic chitosan/DBA core. The result was a high drug loading efficiency of ca. 89.3%, covered by the hydrophilic PEG shells to enhance aqueous stability. In vitro IGC release showed a pH-responsive release, with a cumulative release of IGC of 23% within 1 h at pH 5.0; this compared to 9% at pH 7.4. This was attributed to the cleavage of benzoic-imine bonds at a low pH, leading to structural transition and accelerating the efflux of ICG. Cytotoxicity studies were performed using MCF-7 cells, and the results indicated that PMs were nontoxic. Thus, chitosan-g-mPEG/DBA micelles (CPDMs) are considered a promising pH-responsive delivery platform for cancer theragnostic applications. Likewise, the platform may have potential in dental caries treatment. The hydrophobic chitosan/DBA core could be used to load hydrophobic antibacterial agents such as Far, while the PEG shells could increase the water solubility in oral environments. Moreover, chitosan could be a binding agent targeting the negatively charged surface of bacteria [169] to reduce bacterial adherence [170]. Further, chitosan is a naturally occurring antimicrobial agent used as a mouth rinse for the treatment of oral mucositis [171,172], and thus a synergistic antibacterial therapy for dental caries might be expected when used in combination with other agents.

Apart from the delivery of anticancer agents, chitosan can also be used to coat mesoporous silica nanoparticles (MSNs) as a gate keeper for drug loading. For example, Moorthy et al. (2019) used metal–ligand complex coordination approaches to fabricate MSNs with chitosan oligosaccharide (COS) to form MSNs@COS NPs. These were designed for the pH-responsive drug delivery of DOX (anticancer drug) [173]. DOX loading efficiency was estimated to be around 63%, and the in vitro release showed an obvious pH dependent behaviour, with a negligible amount of Dox release (~5%) observed at pH 7.4 in 24 h compared to 95% at pH 4.0. The pH dependency maybe due to the protonation of the amine and hydroxyl groups under acidic conditions, resulting in electrostatic repulsion and dissociation of the metal–ligand complex [174,175,176,177]. This was then followed by the detachment of the COS polymer layer from the surface of the Dox-MSNs@COS NPs and the subsequent DOX release from the mesoporous channels.

Similar examples can be seen in other fields; Popat’s group (2012) also reported the development of phosphonate-functionalized MSNs (MCM-41-PO^3−^), into which ibuprofen was loaded into pores during a 24 h incubation. These were then capped with chitosan via phosphoramidate chemistry, which is the formation of covalent bonding between the primary amine of chitosan and the phosphonate group on the surface of MSNs. In physiological conditions, chitosan is insoluble and forms a gel-like shell coating the MSN surface; this prevents the ibuprofen being released at pH 7.4. While at a low pH (pH below its isoelectric point at 6.3), amino groups on chitosan were protonated, leading to the swelling of the polymeric matrix and allowing the drug release through diffusion [178].

Nanostructured iron oxide materials such as Fe_3_O_4_ can also be employed as the pH-responsive delivery vehicle. Wu et al. (2010) synthesised hollow Fe_3_O_4_ nanoparticles (NPs) according to the published hydrothermal method [179], and subsequently loaded L-arginine (L-Arg) into the hollow cavity of the NPs via incubation for 24 h. They used the pH-sensitive poly (acrylic acid) (PAA) to coat the NP surface and seal L-Arg inside the Fe_3_O_4_ NPs under neutral conditions (_LP_Fe_3_O_4_ NPs) [180]. The pH-responsive release of L-Arg from _LP_Fe_3_O_4_ NPs was confirmed, with a cumulative release of L-Arg of 72.9% at pH 5.0 in 8h, while only 48.67 and 38.9% was released at pH 6.0 and 7.4, respectively. The low-pH-triggered release behaviour may be due to the protonation of PAA under acidic conditions, which loosens the polymer shell because of electrostatic repulsion, resulting in open pores and L-Arg release. The cytotoxicity of PAA-coated Fe_3_O_4_ (_P_Fe_3_O_4_) and _LP_Fe_3_O_4_ NPs were evaluated by incubating with normal 3T3 fibroblast cells for 24 h, and results showed that neither of the NPs were cytotoxic to 3T3 cells (cell viability above 90% up to 100 μg/mL), indicating the excellent biocompatibility of _P_Fe_3_O_4_ and _LP_Fe_3_O_4_ NPs. Notably, Arginine is a semi-essential amino acid and can be easily accessed via dietary sources [181]. It is also the main component in saliva, and raises the salivary pH by producing alkali in the form of ammonia via the arginine deiminase pathway (ADS) in bacteria. This could be used to neutralize biofilm acidification resulting from bacterial fermentation [182]. Evidence from the past few years suggests that arginine plays an important contribution in reducing biofilm build-up, caries-like lesions and dentinal hypersensitivity [183,184]. In addition, positive responses to arginine treatment were received from people with active caries, as arginine modulates the oral microbiota by normalizing the oral microbiota of the caries-active group to that of caries-free controls. This was determined using typical species abundance, microbial structure and at the transcript level. Furthermore, using arginine in combination with fluoride is of high importance for patients with high caries risk, as the synergistic effect could better promote the growth of alkali-producing Streptococcus sanguinis, while suppressing the acidogenic/aciduric *S. mutans*, significantly attenuating enamel demineralization and delaying the caries progress [185]. Therefore, PAA or modified hollow Fe_3_O_4_ NPs loaded with arginine and fluoride might have promising potential for dental caries treatment.

## 6. Conclusions and Future Perspectives

The human oral cavity consists of approximately 700 identified microbial species (Dewhirst et al., 2010), all of which contribute to the diversity and complexity of the oral microbiota. A sugar-rich diet and/or poor oral hygiene can shift the composition of the oral microbiome and lead to the proliferation of certain cariogenic bacteria that lower the pH and result in dental caries [6]. The use of pH-sensitive nanoparticles for drug delivery have outstanding advantages as the acid sensitivity of the delivery vehicle can respond to cariogenic bacterial species and trigger a controlled release of therapeutic agents in a pathogenic environment without influencing the growth of commensal species. This review summarises pH-responsive delivery vehicles based on their compositions and their mechanisms of pH-responsive drug release. However, compared to the wide range of drug delivery systems (DDS) that have been used for cancer treatment, DDS that target dental caries have been limited to cationic polymeric micelles, silica nanoparticles and chitosan nanoparticles. There are still many areas to be explored, including nanoparticles (liposome, PLGA, dendrimer) and hydrogels (hydroxypropyl methylcellulose (HPMC) and carboxymethyl cellulose (CMC)), which have been used in the pharmaceutical industry with proven safety. Also, to increase the drug loading efficiency, DDS should be carefully chosen, in addition to considering the use of surfactants. Innovative strategies for specific binding to tooth surfaces are also important for DDS, as they not only help to prolong the drug residential time in the oral cavity, but also maximize the antibacterial/antibiofilm efficacy. Although there are still problems that need to be addressed in the development of pH-responsive DDS, considering the advantages in precision drug delivery and the on-demand release of therapeutic agents for enhanced therapeutic impacts, the pH-responsive DDS may prove to be an excellent “adjuvant” for dental caries treatment.

## Figures and Tables

**Figure 1 pharmaceutics-15-01837-f001:**
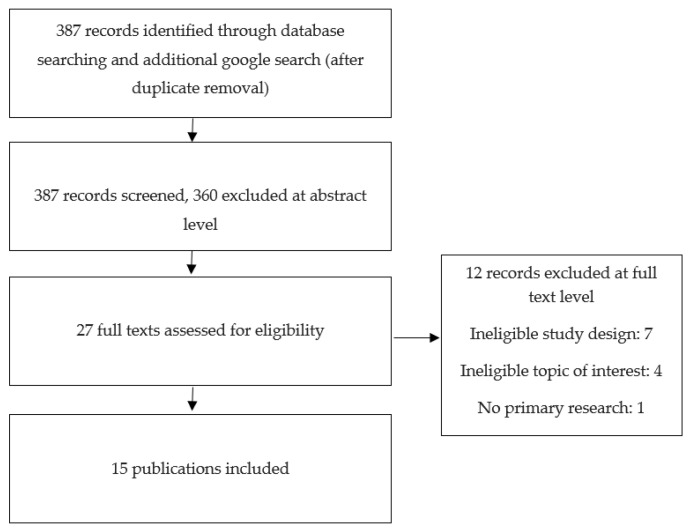
Diagram for the selection of the included publications.

**Figure 2 pharmaceutics-15-01837-f002:**
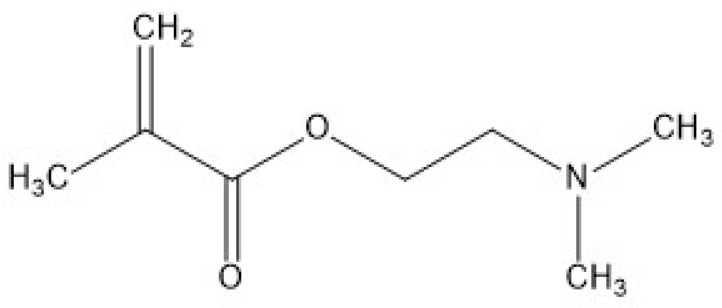
Chemical structure of DMAEMA.

**Figure 3 pharmaceutics-15-01837-f003:**
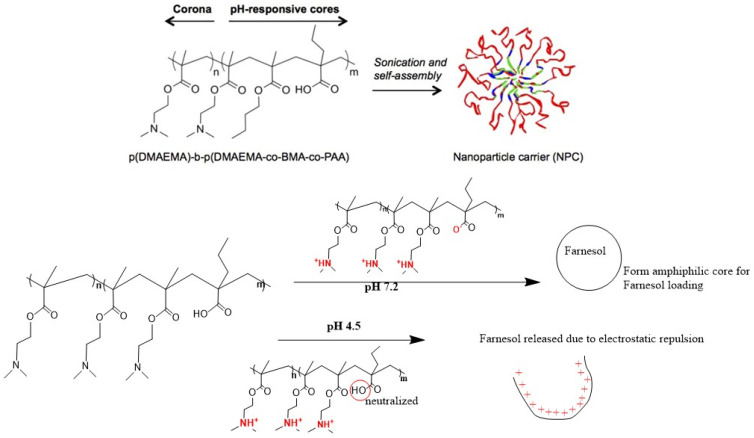
The self-assembly of diblock copolymers and their pH-responsive release mechanisms (the self-assembly component of this figure was reproduced from [30]).

**Figure 4 pharmaceutics-15-01837-f004:**
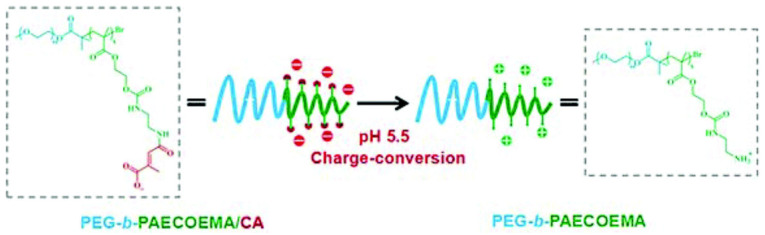
Schematic illustration of the chemical structure of PEG-b-PAECOEMA copolymers and the mechanism of the pH-responsive drug release. PEG-b-PAECOEMA block is negatively charged in neutral conditions. As citraconic amide of PEG-b-PAECOEMA/CA micelles degrades in acidic environments, the primary amine at the end of PEG-b-PAECOEMA becomes protonated and is positively charged [42].

**Figure 5 pharmaceutics-15-01837-f005:**
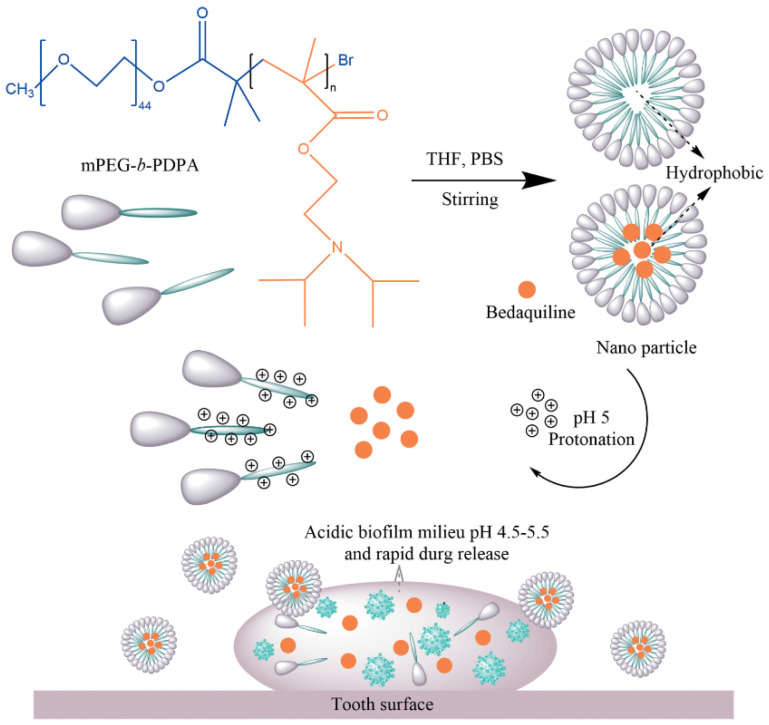
Schematic representation of the formation of mPEG-b-PDPA-based, bedaquiline-loaded micelles and the pH-responsive release mode of bedaquiline from micelles. PDPA block was protonated in acidic conditions and shifted from hydrophobic to hydrophilic, causing the micelles disintegration and drug release [44].

**Figure 6 pharmaceutics-15-01837-f006:**
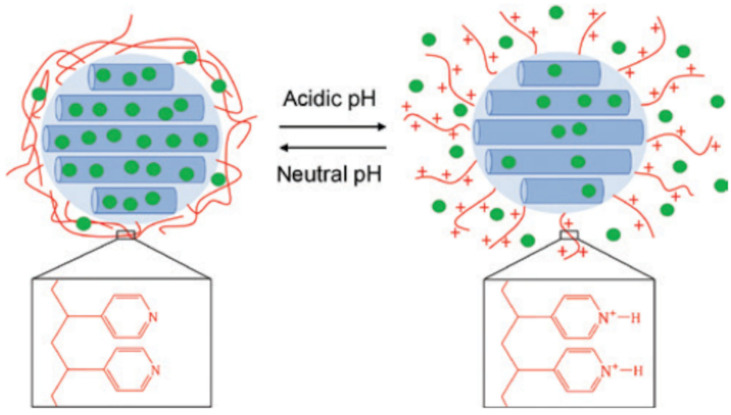
Illustration of the CHX release mechanism from PVP-modified NPSNPs [74]. PVP blocked the nanopores under physiological conditions, preventing CHX release from the carrier, whereas in acidic conditions PVP polymer chains were protonated and straightened up due to electrostatic repulsion and CHX from the reservoir.

**Table 1 pharmaceutics-15-01837-t001:** List of PEG micelles used as pH-responsive drug delivery systems.

Carrier Composition	Therapeutic Agent	Pathogen Used for Antibacterial/Antibiofilm Test	pH-Responsive Release Mechanism	Reference
PEG-b-PAECOEMA/CA	CHX	*S. mutans*	PAECOEMA/CA block is pH-sensitive and the degradation of citraconic amides converts the charge of the carrier from negative to positive, thus positively charged CHX is released via charge repulsion	[42]
mPEG-b-PDPA	Bedaquiline	*S. mutans*	The protonation of the PDPA segment converts the hydrophobic core to hydrophilic when the pH falls below the pKa (pH < 6), causing the dismantling of the nano structure and thus releasing the hydrophobic drug [43]	[44]
PEG-b-PLL/PBA-d.egradable micelles	NaF and TA	*S. mutans*	Catechol groups in TA interact with phenylboronic acid (PBA) in the PEG-b-PLL/PBA blocks and form the pH-sensitive boric acid ester link, which is subject to acid cleavage under cariogenic conditions	[45]
CaCl2 + PEG-Pasp	Doxycycline (DOXY)	*P. intermedia*	DOXY-loaded polymeric PEG-PAsp template-assisted CaCO_3_ mineralized nanoparticles could maintain their mineral structure and keep DOXY from releasing under normal pH in health oral environment, while in acidic pH the DOXY is released due to the dissolution of the CaCO_3_ mineral cores	[46]
PPi-PEGhyd-Far	Farnesal (Far)	*S. mutans*	Via an acid-labile hydrazone bond	[47]

**Table 2 pharmaceutics-15-01837-t002:** List of commonly used pH-responsive drug delivery vehicles in cancer therapy.

Organic NPs	Inorganic NPs
Nanocarrier	Reference	Nanocarrier	Reference
Liposome	[137,138,139,140]	Magnetic nanoparticles	[141,142,143]
Polymeric micelles	[140,144,145,146]	Metal organic frameworks	[147,148,149]
Dendrimers	[150,151]	Carbon nanotubes	[152,153]
Solid lipid nanoparticles	[154,155]	Quantum dots	[150]
Nano-emulsions	[156]	Gold nanoparticles	[157,158]
Hydrogels	[159,160,161]

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
