# Peer review of "pH-Responsive Biomaterials for the Treatment of Dental Caries—A Focussed and Critical Review"

_pharmaceutics, 2023, doi:10.3390/pharmaceutics15071837_

Round 1
Reviewer 1 Report
This paper reviews some pH-responsive nanocarriers and their biofilm targeting mechanisms.
Targeting the local pH environment in caries and therapeutic applications aimed at pH responsiveness would be very useful. However, there are some unclear points in this manuscript as below. Therefore, the reviewer recommends to add further explanation in the manuscript.
If you evaluate its effectiveness as a DDS, it would be better to comment on both the loaded amount and the released amount of the carrier.
I would also like to know how much biofilm was actually destroyed by the technology described in this paper. Please also mention the antimicrobial effect of the nanocarriers.
We also believe that the adsorption properties of biofilms and carriers should be discussed in depth.
In addition, authors should also mention what kind of applications these technologies can be applied to.
The time during which the carrier can exist in the oral cavity is limited. If the carrier cannot selectively adsorb to biofilms and bacteria, it is the same as simply taking nonrestorative therapeutic treatments.
Reviewer 2 Report
The article is relatively well written however a few issues must be addressed. Please see the enclosed Word

A few interpretation and spelling issues were found, English should be revised in the whole document.
Reviewer 3 Report
This manuscript provides an extensive and comprehensive review of potentially caries-inhibiting biomaterials that respond to pH changes and allow for more targeted action. Because these materials are intended to be applied topically rather than systemically, it is perhaps better suited for a dental materials journal than for Pharmaceutics. Nonetheless, the manuscript is well-developed and can be considered part of the broader scope of Pharmaceutics.
Please consider the following minor remarks which may help improve the manuscript:
1. "In Australia, almost 99% of non-fatal burden of disease is related to dental caries [3]." - As the article is aimed at an international readership, it would make sense to refer to the global epidemiology of dental caries rather than focusing on the situation in Australia.
2. Please ensure consistency in terminology. For example, "aciduric" and "acidoduric" are used interchangeably. The former term may be more common.
3. "An additional Google search was performed using the above-mentioned selection criteria." - Please indicate whether this was a Google Scholar search or a plain Google search.
4. "DMAEMA (dimethylaminoethyl methacrylate) is a cationic polymer..." - Is this not a monomer? It only becomes a polymer after polymerization...
5. Also, generic drug names should be lowercase (Farnesol and farnesol are used interchangeably in the text).
6. "CHX is considered a gold standard in dental clinical practice and has been widely used for its broadspectrum antibacterial effects [31]." - This is true, but only for certain indications (e.g., mouth rinses). For some other purposes (e.g., endodontic irrigation), CHX is not a gold standard. Please clarify this to avoid misunderstandings.
7. It appears that different citation styles are being used interchangeably. For example:
Zhao et al [41] synthesised (...)
Zhang et al (2021) used mPEG-b-PDPA (...)
Please check and revise the consistency of citations.
8. Some figures are identical to previously published material. For example, Figure 4 is identical to "Scheme 1" in the following article: 10.3389/fmicb.2021.761583
On the other hand, the authors clearly stated in the caption of Figure 2 that "the self-assembly component of this figure was reproduced from [29])." However, no such statement is provided for Figure 4.
Please make sure you have the necessary permissions to reuse figures.
9. Please thoroughly recheck the manuscript for typos, double spaces, and unnecessary characters. For example, the sentence "The biofilm extracellular polymeric matrix (provides protection to bacteria from environmental changes such as pH, osmolarity, mechanical and shear forces [77-79]." contains an unnecessary open parenthesis.
10. Check the reference list, for example: [160] (!!! INVALID CITATION!!! [130-132]).
The use of English is fine, but please check for multiple typos.
Round 2
Reviewer 1 Report
出版が受理されることをお勧めします。
Reviewer 2 Report
The manuscript has been improved